# Convergent evolution and B-cell recirculation in germinal centers in a human lymph node

Aurelien Pelissier[1,2,*], Maria Stratigopoulou[3,*], Naomi Donner[3], Evangelos Dimitriadis[4], Richard J Bende[3], Jeroen E Guikema[3,†], Maria Rodriguez Martinez[1,†], Carel JM van Noesel[3,†]

Germinal centers (GCs) play a central role in generating an effective immune response against infectious pathogens, and failures in their regulating mechanisms can lead to the development of autoimmune diseases and cancer. Although previous works study experimental systems of the immune response with mouse models that are immunized with specific antigens, our study focused on a real-life situation, with an ongoing GC response in a human lymph node (LN) involving multiple asynchronized GCs reacting simultaneously to unknown antigens. We combined laser capture microdissection of individual GCs from human LN with next-generation repertoire sequencing to characterize individual GCs as distinct evolutionary spaces. In line with well-characterized GC responses in mice, elicited by immunization with model antigens, we observe a heterogeneous clonal diversity across individual GCs from the same human LN. Still, we identify shared clones in several individual GCs, and phylogenetic tree analysis combined with paratope modeling suggest the re-engagement and rediversification of B-cell clones across GCs and expanded clones exhibiting shared antigen responses across distinct GCs, indicating convergent evolution of the GCs.

## Introduction

Germinal centers (GCs) are specialized microanatomical structures within the secondary lymphoid organs where B cells proliferate, differentiate, and mutate their antibody genes in response to the presence of foreign antigens (1, 2). Through the GC lifespan, interclonal competition between B cells leads to an increased affinity of the B-cell receptors (BCRs) for antigens accompanied by a loss of clonal diversity in rodents (3) and humans (4). Throughout the GC reaction, B cells with improved affinity for antigens as a result of these mutations are continuously selected. By competing for antibody-mediated antigen capture and subsequent acquisition

of T-cell help, the mutated B cells gain affinity, and the selected B cells with improved affinity for the antigen differentiate into memory B cells and plasma cells (1, 5). GCs thus play a central role in generating an effective immune response against infectious pathogens, and failures within their tightly regulated environment can lead to the development of autoimmune diseases (6) and cancer (7).

In vivo mouse experiments have shown that individual GCs exhibit variable degrees of clonal imbalance and dominance, even when induced synchronously by immunization with various antigens (CGG, chicken γ globulin; OVA, chicken ovalbumin; HA, influenza hemagglutinin (H3); NP-OVA, 4-hydroxy-3-nitrophenylacetyl-OVA) (3). In particular, a subset of GCs underwent a massive expansion of higher affinity B-cell variants (clonal bursts), leading to a loss of clonal diversity at a significantly faster rate than other GCs. These differences in clonal dynamics among GCs could partially be explained by the differences not only in affinity between competing B cells but also by other factors unrelated to affinity (3). Moreover, the temporal resolution of GC reactions through intravital microscopy revealed substantial heterogeneity in the evolution of both foreign antigen induced and autoreactive GCs over time (8). Interestingly, initially, dominant clones were observed to suddenly lose competitive momentum allowing for the emergence of other clones (inversion event) (8). GC computational models suggest that these chaotic dynamics were likely the consequences of initially small stochastic advantages in the affinity to antigens (9), amplified through the selection and proliferation of higher affinity clones. A research question of interest is to which extent these observations are applicable to a typical ongoing GC response in human LN, which involves multiple asynchronized GCs to unknown antigens.

During lymphocyte development, the antigen receptors expressed by B and T lymphocytes are assembled in an antigen-independent fashion by ordered variable gene segment recombination (V(D)J recombination) (10). The antigen receptors of B cells are then further diversified in the GCs through somatic hyper mutation (SHM) (11) which is induced by activation-induced cytidine deaminase (AID),

[1]IBM Research Europe, Rüschlikon, Switzerland   [2]Department of Biosystems Science and Engineering, ETH Zurich, Basel, Switzerland   [3]Department of Pathology, Amsterdam University Medical Centers, Location AMC, Lymphoma and Myeloma Center Amsterdam, Amsterdam, Netherlands   [4]MonetDB Solutions, Amsterdam, Netherlands

Correspondence: j.e.guikema@amsterdamumc.nl; mrm@zurich.ibm.com; c.j.vannoesel@amsterdamumc.nl
*Aurelien Pelissier and Maria Stratigopoulou contributed equally to this work
†Jeroen E Guikema, Rodriguez Martinez, and Carel JM van Noesel contributed equally to this work

both base excision repair and mismatch repair are necessary for those processes (12, 13). Affinity maturation through SHM is critical for antibodies to reach high affinity for the target antigen (11). During V(D)J recombination and SHM, stop codons or frameshifts of the BCR sequences may occur, leading to non-functional BCR sequences. GC B cells that concur SHM-derived crippling mutations, resulting in failure to express a functional BCR, undergo apoptosis (14). On the other hand, non-functional BCR sequences derived from V(D)J recombination may be propagated in the GC but are not subjected to selection (1). A recent study focusing on mice Peyer patches (15) sequenced both functional and non-functional V(D)J rearrangements from genomic DNA (gDNA), which they used to identify positively selected mutations by comparing SHM mutation rates between functional and non-functional sequences. SHM selection was identified for some shared clonotypes among mice under different gnotobiotic conditions, but not for all of them (15).

In vivo studies in mice have revealed that recently activated B and T cells could constantly enter ongoing GC reactions (16, 17, 18), thus indicating that GC seeding is an ongoing process governed by a competitive advantage in antigen-binding affinity of naive B cells (2, 16). In addition, in vivo time resolution of single GCs in mouse models demonstrated a possible reentry of output memory B-cell clones to ongoing GCs (8). In line with these results in mice, we demonstrated reoccurring B-cell clones in multiple GCs within the human LNs (19), suggesting the migration of antigen-responsive B cells in human lymphoid tissue. Those findings support the hypothesis that high affinity BCRs do not necessarily arise during a single GC passage, but potentially after successive GC responses (19). Nevertheless, such phenomena were shown to be infrequent in mouse models under typical boost regimens (20), with less than 10% of the clones found in multiple GCs, and secondary GCs consisting predominantly of B cells without prior GC experience.

Importantly, although mouse immunization models represent controlled environments in which B cells from all GCs react to the same antigen, a major question is to which extent observations in such experimental models are applicable to a typical ongoing GC response in a human LN that involves multiple asynchronized GCs reacting simultaneously to unknown antigens.

We have developed a method to study the BCR repertoire at the GC scale by combining LCM of individual GCs from human LNs with next-generation sequencing (NGS)-based immunoglobulin heavy chain variable region (IGHV) repertoire analysis (Rep-seq) from gDNA. Our study goes beyond most of the previous GC mouse studies (8, 20) in several aspects. Firstly, the use of gDNA, as opposed to more commonly used RNA, gives a direct and unbiased reflection of clonal frequency because one V(D)J rearrangement equals one B cell, which is not the case with mRNA analysis. Secondly, it allows the analysis of the source and the fate of the non-functional BCR sequences. Furthermore, it enables the analysis of the effects of affinity-based selection on the specific mutational spectrum during SHM, at the individual GC scale. Such analyses were performed previously in mouse models in bulk naive and GC B cells from the spleen or Peyer's patches (15), but not in individual GCs. Importantly, the use of NGS-based Rep-seq of individual GCs enables the sequencing of more than $10^4$ B cells in each GC, which considerably strengthens the quantitative and statistical analysis of the GC repertoires, with regards to clonal

representation and diversity. Finally, phylogenetic analysis provide insights into the functional convergence of dominant clones across GCs by identifying shared clones, which represent reactivated B cells and expanded clones across distinct GCs.

# Results

## Heterogeneous clonal diversity across GCs

We performed LCM on 10 individual GCs from a human LN via using five serial tissue sections (Fig S10) and we applied Rep-seq to obtain sequencing data of the IGHV of the BCR, obtaining ~100,000 sequences per sample (Fig 1A and E). Two replicate PCR amplifications were performed for each GC and analyzed to verify the reliability and reproducibility of our PCR amplification and NGS approach, and to quantify the variance intrinsic to our measurement protocol, as multiplex PCR approaches may be subject to amplification bias, especially when PCR substrate is limiting. To that end, we studied the effect of the DNA concentration on PCR reproducibility by comparing the replicate analysis performed on only one isolated tissue section per GC with the replicate analysis from five (serial) tissue sections per GC. We observed much lower Sørensen–Dice similarity indices when we used only one tissue section, indicating variegated bias in each individual amplification and sequence run when PCR substrate is limiting (Section 1 in Supplemental Data 1). However, when using ~fivefold more PCR input, replicate analyses showed excellent concordance (Fig 1B), suggesting that PCR bias has a minor impact on our results. Moreover, mutation analysis revealed that all samples were very similar in terms of mutation rate and the nature of these mutations (Section 2.1 in Supplemental Data 1). Following previous conventions (3, 15, 21 Preprint), sequences were grouped together into clones by shared V, J gene segments and CDR3 length, and more than 84% junction nucleotide sequence identity, as optimized from the distance to the nearest distribution model (22) (Section 3 in Supplemental Data 1). We analyzed the similarity between the clonal repertoire of each sample and observed a high similarity between samples from the same GC (Fig 1B and C), confirming the reliability of our sequencing protocol, showing a low overlap between IGHV sequences obtained from different GCs (<0.1), suggesting that GCs are distinct evolutionary environments and thus relatively independent from each other.

We then classified sequences into three categories according to their frequency, corresponding to the dominant clone (i.e., most abundant clone), an expanded clone (frequency >1%), or a non-expanded clone (frequency <1%, typically involving less than 1,000 sequences belonging to that clone in our data). As highlighted in Fig 1D, the clonal abundance within different GCs is heterogeneous.

To quantify the heterogeneity between different GCs, we computed the clonal diversity in terms of dominance (proportion of the most abundant clone), evenness (homogeneity of the clonal abundance), richness (number of clones), and Shannon entropy, an alternative measure of diversity, that is, less sensitive to singletons (clonal group with a unique sequence). In agreement with previous mouse models (3, 8) that identified heterogeneous clonal dominance across GCs, we

**Figure 1. Sample diversity analysis.**
**(A)** Experimental framework combining laser capture microdissection (LCM) and Rep-seq from human lymph node (LN) GC's genomic DNA to analyze F (functional) and NF (non-functional) rearrangements via semi-nested PCR amplifying the leader (LD) and framework region 1. Details of the experimental approach are provided in the methods section. The area within the dashed lines corresponds to the area that was isolated by LCM and used for genomic DNA extraction. **(B)** H&E staining performed in parallel with LCM is shown (10X magnification) (B) Sørensen–Dice similarity between each sample in terms of clonal abundance. **(C)** Clone abundance across samples, denoted as Vgene_JunctionLength_Jgene (only the 20 most abundant clones are shown in the legend for visual clarity). **(D)** Proportion of sequences belonging to the dominant clone, expanded clones, and non-expanded clones in each sample. **(E)** Number of sequencing reads in each sample (in units of 1,000). **(F)** Diversity analysis across samples in terms of dominance, richness, Shannon entropy, and evenness, where $^q D$ corresponds to Hill's unified notation. To highlight the relevance of studying GCs individually, each sample (blue) is compared with an artificial sample of equivalent size sub-sampled from combining all the obtained sequences (grey).

show that the clonal diversity within each GC takes a wide range of values (Fig 1C and F). As an example in our setting, the clonal dominance ranged from 5% (GC9) to 30% (GC10). To study the role of sample size in the GC heterogeneity quantification, we paired each sample with an artificial sample of equivalent size, obtained from randomly selecting sequences from all the GCs combined. As highlighted in Fig 1F, differences related to sample size are not statistically significant compared with the diversity variation across samples. These findings highlight the importance of studying individual GCs and prove that the heterogeneity is not a result of sub-sampling. In Fig S11A–D, we show that the variability in the diversity metrics between samples is consistent across different clonal identification methods, where the same conclusion can be obtained when quantifying diversity in terms of CDR3 abundance only.

### Heterogeneous V gene repertoire usage across GCs

Analyzing the abundance of V genes in each sample reveals that GCs differ significantly in terms of their V gene usage (Fig 2A–C and Section 4 in Supplemental Data 1). Interestingly, three genes in particular stand out with regards to their frequency across all

samples: IGHV1–2, IGHV2–5, and IGHV1–18 (Fig 2E). Although these genes were shown to be the most frequently found in the peripheral blood of adult patients suffering from end-stage renal disease (23), they were still found in a higher abundance in our data than in other LNs and the bone marrow of healthy donors analyzed by a BIOMED-2–based multiplex PCR amplification and sequencing approach similar to ours (24) (Fig 2E), suggesting that these V genes may have been positively selected during the GC reaction in the reactive lymph node used in our study. This is further supported by the fact that these genes are found at a higher frequency than the most abundant V genes in non-functional BCR sequences (Fig 2F), which are not subjected to selection. The same figure with functional and non-functional V gene abundances for the same V gene label is provided Fig S13.

Next, we analyzed the V gene usage in the top 15 dominant clones from each individual GC. We found that the frequencies of the IGHV1–2, IGHV2–5, and IGHV1–18 genes varied significantly between the GCs (Fig 2D). As an example, in GC6, the IGHV1–18 or IGHV2–5 genes were not found in the top 15 clones, whereas they were present in seven and five of the top 15 clones from GC9,

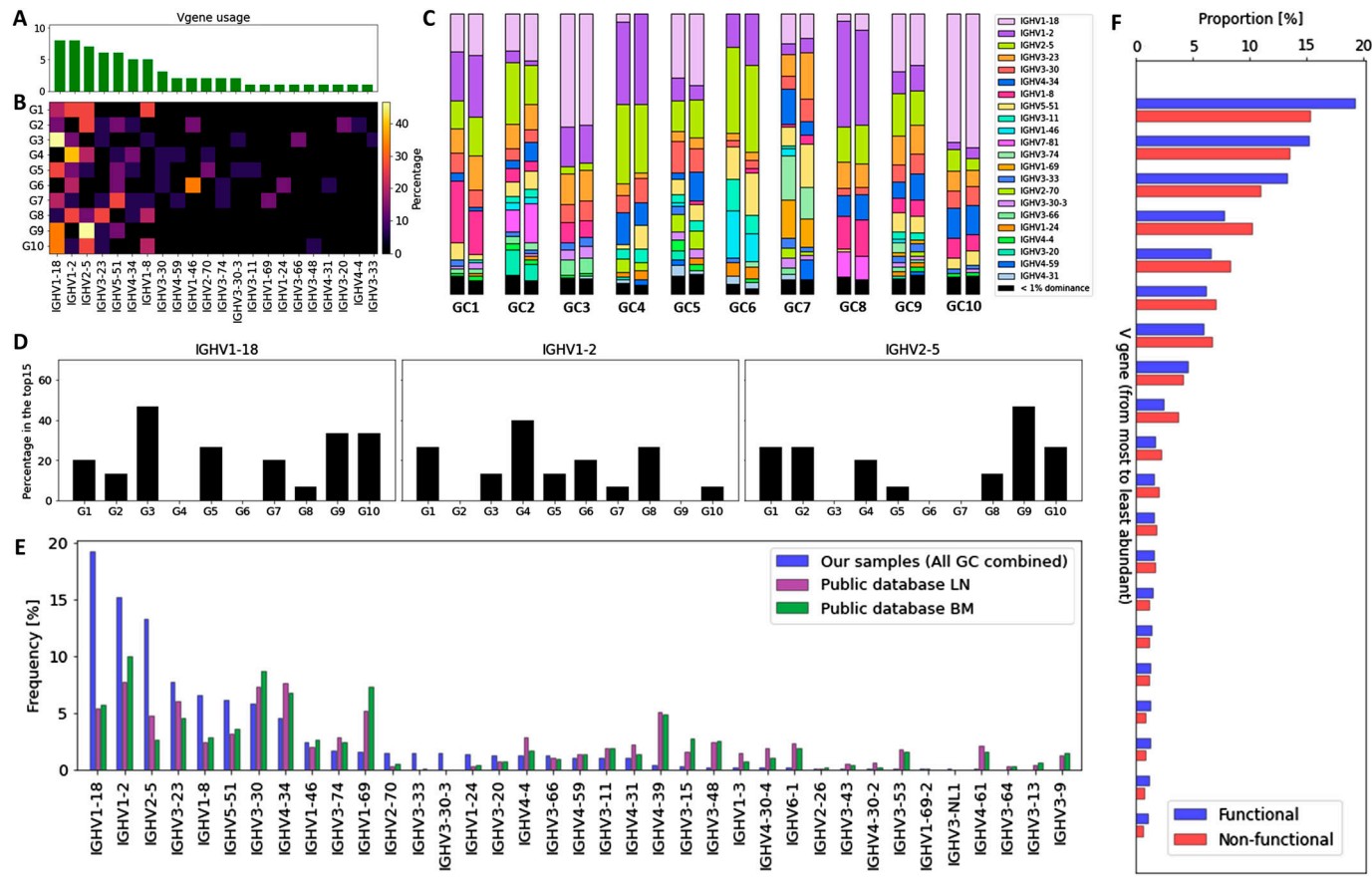

**Figure 2. V gene repertoire usage across samples.**
**(A)** Histogram expressing the number of GCs for which a given V gene is used by the 15 most abundant clones of that GC, ranked from most to least abundant.
**(B)** Heatmap where a pixel represents the frequency (in %) with which a given V gene has been observed in a given GC's 15 most abundant clones. **(C)** V gene abundance across samples. **(D)** Histogram showing the V gene usage by the 15 most abundant clones in each GC. The three most abundant V genes (IGHV1–18, IGHV1–2, IGHV2–5) are depicted. **(E)** Comparison of V genes frequency in our samples (all GCs combined) and in B cells from a public dataset (24). **(F)** V genes frequencies in functional and non-functional B cells, ranked from the most to the least abundant. The V gene labels are not the same for non-functional and functional sequences, as we are only interested in comparing the shape of the V gene distribution. The same figure with common labels is provided as Fig S12.

respectively. Similar results were obtained when using different thresholds for dominance (Section 4 in Supplemental Data 1). Moreover, we analyzed the clones most abundantly encountered in our analysis and their features with two different sequence-grouping algorithms. We used the TRIP tool (25) to group the sequences with identical CDR3 amino acid sequences into clones. In parallel, we used the hierarchical agglomerative clustering algorithm to cluster the junctional sequences (26 *Preprint*), which was previously proven to be effective at identifying clones (27). This approach requires identical V and J gene segments, identical CDR3 lengths, and more than 84% junction nucleotide sequence overlap to assign two sequences to the same clone (27) (see the Material and Methods Section "Grouping clone sequences" for more details). Both algorithms result in a similar identification of the most abundant clones. We observed that the clones most abundantly encountered in our analysis exhibit common features in terms of CDR3 length and V, D, and J gene usage (Section 5 in Supplemental Data 1). These data reveal different V genes expansions in individual

GCs and convergent evolution of the expanded and most abundant clones.

### Heterogeneous clone functionality across GCs and SHM-induced crippling mutations

We classified the sequences into four categories based on the identification of a frameshift or a stop codon. The categories are the following: (i) functional sequences, (ii) out of frame non-functional and in frame non-functional due to an early stop codon induced by (iii) V(D)J recombination or (iv) SHM. Out of frame sequences were defined as having an out of frame junction, with a frameshift detected by IMGT-V-Quest (28). In frame non-functional sequences were assumed to arise either from V(D)J recombination when having at least one stop codon in the IMGT N-region or from SHM otherwise. More specifically, the BCR junction is formed when the germline V, D, and J genes are associated during V(D)J recombination. During this process, additional nucleotides are inserted between the V, D and D, J genes, which are referred to as the

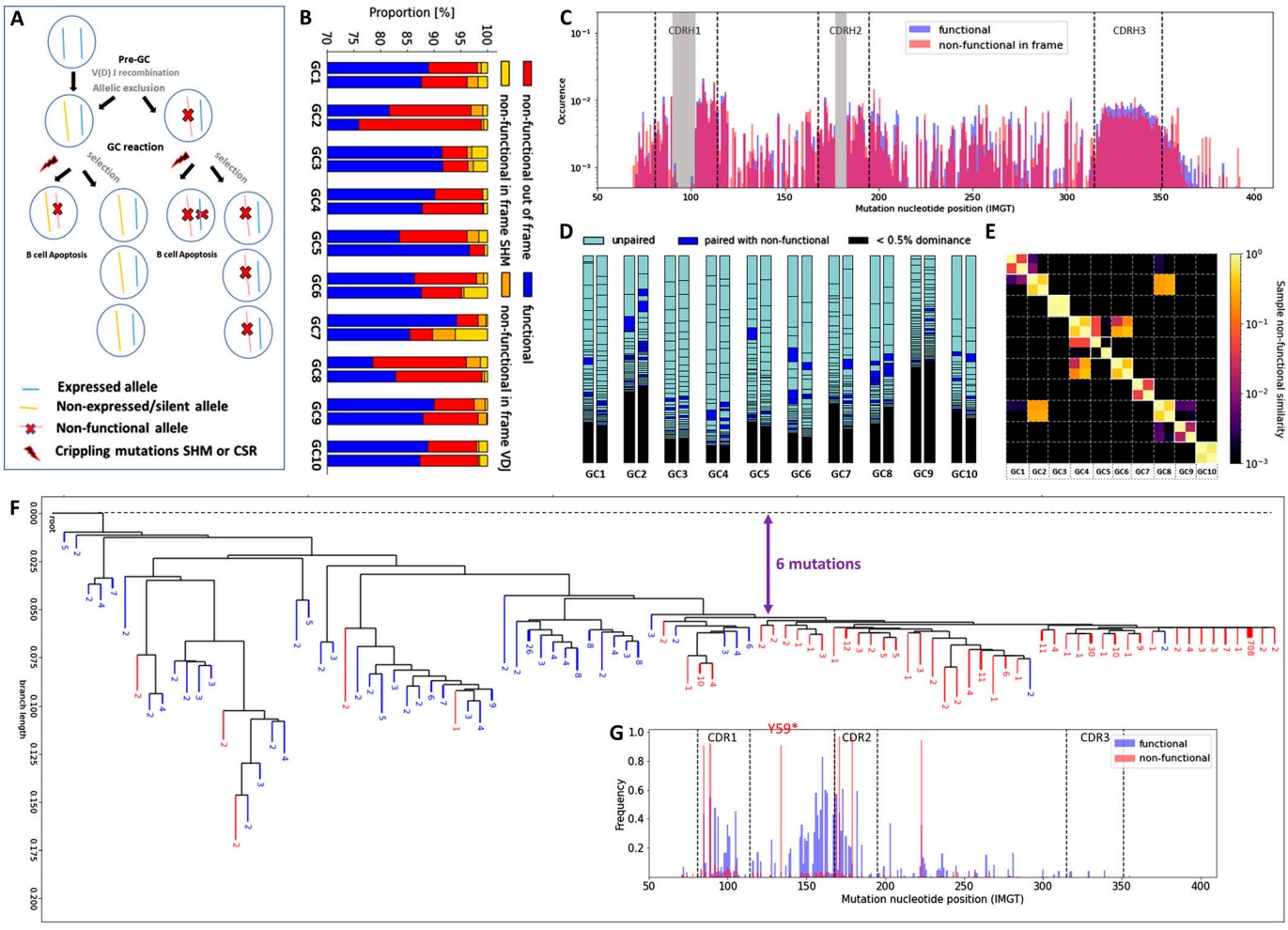

**Figure 3. Analysis of functional and non-functional BCR sequences in GCs.**
**(A, B)** Diagram showing the possible fates of functional and non-functional B-cell alleles before and during a GC reaction, (B) proportion of functional and non-functional in frame and out of frame sequences in each sample. **(C)** Mutational spectrum regrouping the position-wise mutation frequency across all sequences. The near zero mutation frequency zones in the middle of CDR1 and CDR2, shown in grey, are due to gaps related to the IMGT numbering scheme that varies across sequences. Mutations were inferred by comparison to the most common sequence segments among all sequences within a given clone. **(D)** Mapping of functional and non-functional clones across samples. Each rectangle represents a functional clone. Dark blue rectangles represent functional alleles paired with non-functional alleles, whereas light blue rectangles represent unpaired functional alleles. **(E)** Sample similarity in terms of frequency of non-functional clones. **(F, G)** Phylogeny reconstruction and (G) mutational spectrum of a representative clone including both functional and non-functional alleles (IGHV3–74_60_IGHJ5). Mutations associated with functional sequences are depicted in blue, whereas non-functional ones are shown in red. The crippling mutation Y59* was probably caused by SHM and made the BCRs non-functional. The distance between most of the non-functional BCR sequences and the root can be quantified in terms of number of mutations, highlighted in purple. The analysis of other F and NF clones is provided in Fig S13.

N-regions (29). Consequently, we assumed that stop codons in N-regions are mostly derived from V(D)J recombination errors.

We observe that most of the sequences is functional (≈90%), and 78% of the non-functional sequences result from a frameshift. The remaining sequences are roughly equally divided between SHM-induced and V(D)J recombination–related stop codons. The distribution appears consistent between GCs, with the proportion of non-functional sequences ranging between 10% and 25% across samples (Fig 3B). The frequency of non-functional sequences was found to be relatively similar across samples from the same GC, except for GC5, where non-functional sequences were found in a low abundance in one of the two NGS samples. The high degree of similarity of the non-functional sequences in GCs 4–6 and GCs 2–8

(Fig 3E) was related to shared clones in these GCs, which also carried the non-functional allele, supporting the notion that these are due to V(D)J recombination errors. Studies in mouse models focusing on the role of crippling mutations during SHM identified a limited number of stop codons in the GC with a range of 2–5%, where the crippling mutations were strikingly enriched in B cells, expressing low surface BCR levels, being seven times more abundant (17.2%) (14) compared with our current analysis of a human lymph node.

We computed the SHM mutational spectrum by regrouping the position-wise mutation frequencies across all sequences. Our data highlighted an expected increase in the mutation frequency in the CDR regions, but no significant differences between functional and

non-functional sequences were found in this context (Fig 3C). The same comparison cannot be performed at the clonal level because each clone has different V or J genes and thus the mutation positions cannot be compared across different clones. Furthermore, we also studied the SHM spectrum with regards to the type of mutations induced by AID in each GC for the functional and the non-functional clones. Mutation analysis revealed that all samples were very similar in terms of mutation rate and the nature of these mutations (Fig S2A). Interestingly, no difference between functional and non-functional clones regarding the SHM spectrum was observed suggesting that the SHM process seems to be stochastic. However, the selection pressure of the functional dominant clones was assessed by studying the replacement to silent mutation (R/S) ratio in CDR for the functional expanded clones versus the singletons (Fig S2B). The R/S ratio indicates that replacement mutations are being selected over silent mutations and it positively correlates to the selection pressure undergoing in a GC environment. We observed increased R/S ratio for the functional dominant clones compared with the "singleton" sequences, verifying the difference of selection pressure between the expanded and the non-expanded clones (Fig S2C).

## Pairing between functional and non-functional alleles

The expansion of non-functional IGH alleles in the GC may be derived from selected B cells that harbor both a functional and a non-functional allele due to V(D)J recombination errors (30, 31) (Fig 3A). Based on this notion, we paired each non-functional IGH allele with a functional IGH allele by assuming that equal/similar abundance indicates that they come from the same B cell. We show in the obtained mapping that the paired functional and non-functional alleles were observed in all the GCs, consistently in both replicates. Furthermore, the frequency of the functional alleles that were paired with non-functional ones varies between individual GCs (Fig 3D).

## Identifying non-functional clones derived from SHM

We performed functionality analysis also at clonal level across GCs and separated the clones into three categories. We defined a clone as functional if it contained more than 95% functional sequences (F), non-functional (NF) if it contained less than 5%, and F and NF otherwise, that is, clones consisting of both functional and non-functional sequences. The F and NF clones corresponded to the 4% of all the clones studied, without significant correlation with their abundance (i.e., being F and NF seems to be independent of being selected). All of them were in frame and likely caused by SHM because the stop codons were outside of the junction region. We found at least 5 F and NF clones with 1,000 different unique sequences and investigated these in more detail. After inference of their phylogenetic trees, we can observe a separation between functional and non-functional branches (Figs 3F and S13). Thus, we can check for mutations that were selected through affinity maturation, as performed previously in mouse models (15). For example, we observe several selected mutations of the functional sequences in the FWR2 region of the studied F and NF clone (Fig 3G). Still, we did not identify specific mutations consistently selected across clones (Fig S13). On the other hand, our analysis revealed

that specific crippling mutations occurred independently in different F and NF clones, when they shared the same V genes (Table S1). As an example, the mutation Y59* occurred consistently in five independent clones with the IGHV3 gene.

Regarding the timing of these crippling mutations, the distance between the non-functional sequences and the root in the inferred tree can inform us about the number of mutations the BCR sequences underwent before the crippling mutation occurred, which correlates with the number of cell divisions (roughly one mutation per two cell divisions (9)). It was estimated from our generated trees that an average of 3.6 mutations occurred before the crippling mutation (Fig S13).

Expansions of specific V genes are frequently involved in stereotypic rearrangements found in B-cell malignancies (IGHV1–8, IGHV1–2, IGHV3–23, IGHV4–34 (32, 33)), autoimmune diseases, and infectious diseases (IGHV4–34, IGHV5–51, IGHV1–69, IGHV1–46 (34)). We observed that most of the F and NF clones with crippling mutations (22/35) use some of those genes. More specifically, the genes IGHV1–2, IGHV3–23, IGHV4–34, and IGHV2–5 were used by the F and NF clones. The SHM can act as a double-edged sword for the organism because it is necessary for an effective immune response, but at the same time it introduces mutations that can induce the recognition of self-antigens and consequently can lead to autoimmune disease. The GCs not only rely on the selection of the antibodies with the highest affinity but also on autoreactivity checkpoints that are needed for the counter-selection of B cells that can bind to self-antigens (1). As studies in mouse models demonstrated that self-reactive GC B cells are counter-selected or inactivated by SHM (35), our finding supports this hypothesis of counter-selection mechanism of self-reactive B cells in the human GC. Still, the role of negative selection in the GC is not yet completely clear, despite the deepened knowledge obtained from recent studies using mouse models (14, 36).

## B-cell reactivation in the GC

Several studies in both rodents and humans by us and others have shown various frequencies of shared clones between different GCs (3, 19, 20), which might be indicative of B-cell reactivation in ongoing GC reactions (8). We investigated the number of shared clones in a human LN. To increase the robustness of our analysis, we considered a clone to belong to a GC only if it was consistently identified in both NGS replicates. We found 10.8% (396/3,650) of functional clones to be present in at least two GCs (Fig 4B). This is in line with a recent study in the mouse, where ~10% of the clones were found to be shared between individual GCs induced by CGG immunization (20). Moreover, some of the shared clones exhibit common features with other shared clones across GCs, regarding V, D, and J gene usage and CDR3 length (Section 5 in Supplemental Data 1). Shared clones between different GCs indicate a frequent B-cell recirculation from one GC to another (Fig 4A). Nevertheless, clones that were expanded in more than two GCs were rare and most of the shared clones were expanded only in one GC, with only 5% of clones found with dominance >0.1% in more than one GC (Fig 4B). The fact that shared clones exhibit different degrees of dominance in different GCs fit with the notion that GCs are distinct competitive environments, leading to clonal expansions varying by several orders of magnitude across GCs. In fact, a recirculating B cell

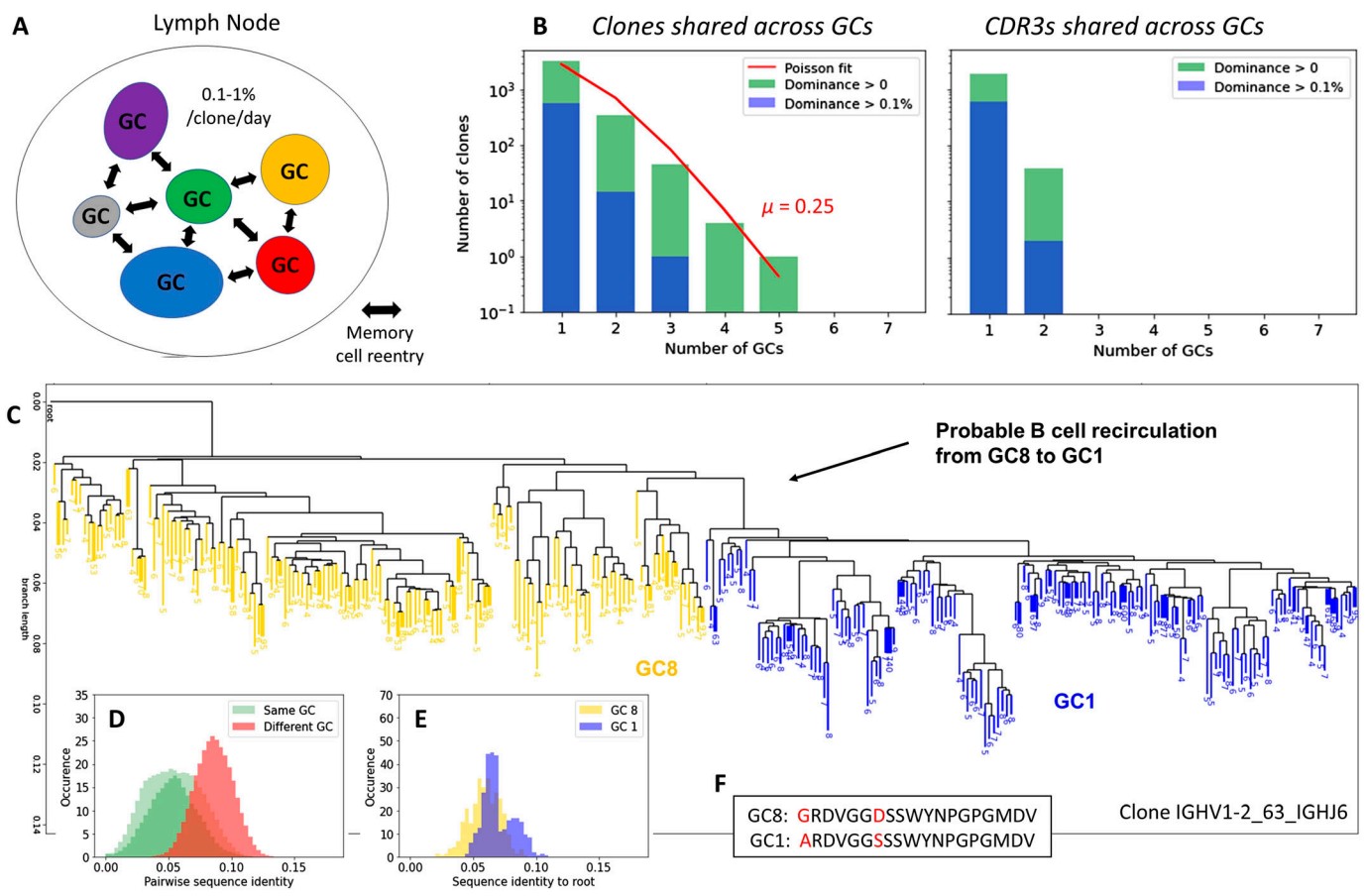

**Figure 4. B-cell recirculation across Germinal centers.**
**(A)** Cartoon representing the probability of clones being shared between different GCs of the same LN. **(B)** Number of clones and number of identical CDR3 sequences as a function of the number of GCs in which they are found (in green), fitted to a Poisson process (in red) of parameter rate $\mu$. The blue histogram depicts the clones that are found with at least 0.1% dominance. **(C)** Phylogenetic reconstruction of clone IGHV1–2_63_IGHJ6, where each colored branch corresponds to a unique BCR sequence. The number of replicates of each BCR sequence is indicated at the bottom of the branch. Only the 150 most abundant sequences from both GC8 and GC1 are shown for visualization clarity. **(D)** Histograms of the pairwise sequence similarity between BCRs of the same GC (green) and different GCs (red). **(E)** Histograms of the pairwise sequence similarity between BCRs and the root of the joint tree including both GC1 and GC8. **(F)** Most common CDR3 sequences among BCRs from both GCs.

enters different GCs at a disadvantage compared with other already established dominant clones, where its initially low abundance makes it more difficult to compete for T cell help (9).

Interestingly, the distribution of shared clones among GCs fits a Poisson distribution well (Fig 4B), suggesting that the reactivation of a B-cell in a different GC is a memoryless stochastic process where the past evolutionary history of the B cell does not play any role. According to the Poisson distribution and denoting as $\lambda$ the occurrence rate (in our case the rate of B-cell reactivation), the number of occurrences $N$ at the exposure time $t$ is a random variable of mean $\mu = \lambda t$ and distribution:

$$P[N = n] = \frac{\mu^n e^{-\mu}}{n!}. \tag{1}$$

Fitting Eq. (1) to the observed distribution gives $\mu = 0.25$. From this, we can compute the probability that a given clone seeds at least one other GC during the whole GC reaction $P(N \geq 1) = P(N \geq 0) - P(N = 0) = 1 - e^{-\mu} = 1 - e^{-0.25} \approx 22\%$.

We can also estimate the probability of seeding per clone and day. When using 20 d as a typical lifetime of a GC, generated in response to immunization with a simple hapten-protein conjugate such as NP-CGG (3), we can estimate the GC B-cell reactivation rate to be $\lambda \approx \frac{0.25}{20} = 0.0125$ seeding per day per clone. However, it is worth keeping in mind that the calculated reactivation rate is only an upper limit, as the GC lifetime most likely is much longer in chronically activated LNs. For instance, lifetimes of up to 100 d have been observed for some viral infections (37), and GCs formed in response to synthetic antigens can persist for up to 1.5 yr (38). Such extended GC lifetimes would result in much smaller reactivation rates. As an example, if the data would have been collected at day 200 of the GC reaction, the estimated reactivation rate would decrease to 0.1% (Fig 4A). We are interested not only in how many clones (sequences sharing the same V, J gene segments and CDR3 length, and more than 84% junction nucleotide sequence identity) are shared across GCs but also in how many identical CDR3s are observed between different GCs. We observed that only 2% (38/1,885) of the CDR3s are shared among GCs, which

is considerably less than for clones. This indicates that the CDR3 typically continues to mutate after B-cell recirculation in the new GC environment (Fig 4F). To gain insight into the parallel evolution of a given clone in different GCs, we inferred phylogenetic trees from clones found in multiple GCs. In Fig 4C, we show the

phylogenetic tree inferred from clone IGHV1-2_63_IGHJ6, which was found expanded with dominance >1% in two different GCs. The hierarchical reconstruction of the tree clearly separated the two GCs, where sequences in the same GC were more similar to each other than sequences in different GCs (Fig 4D). In this example,

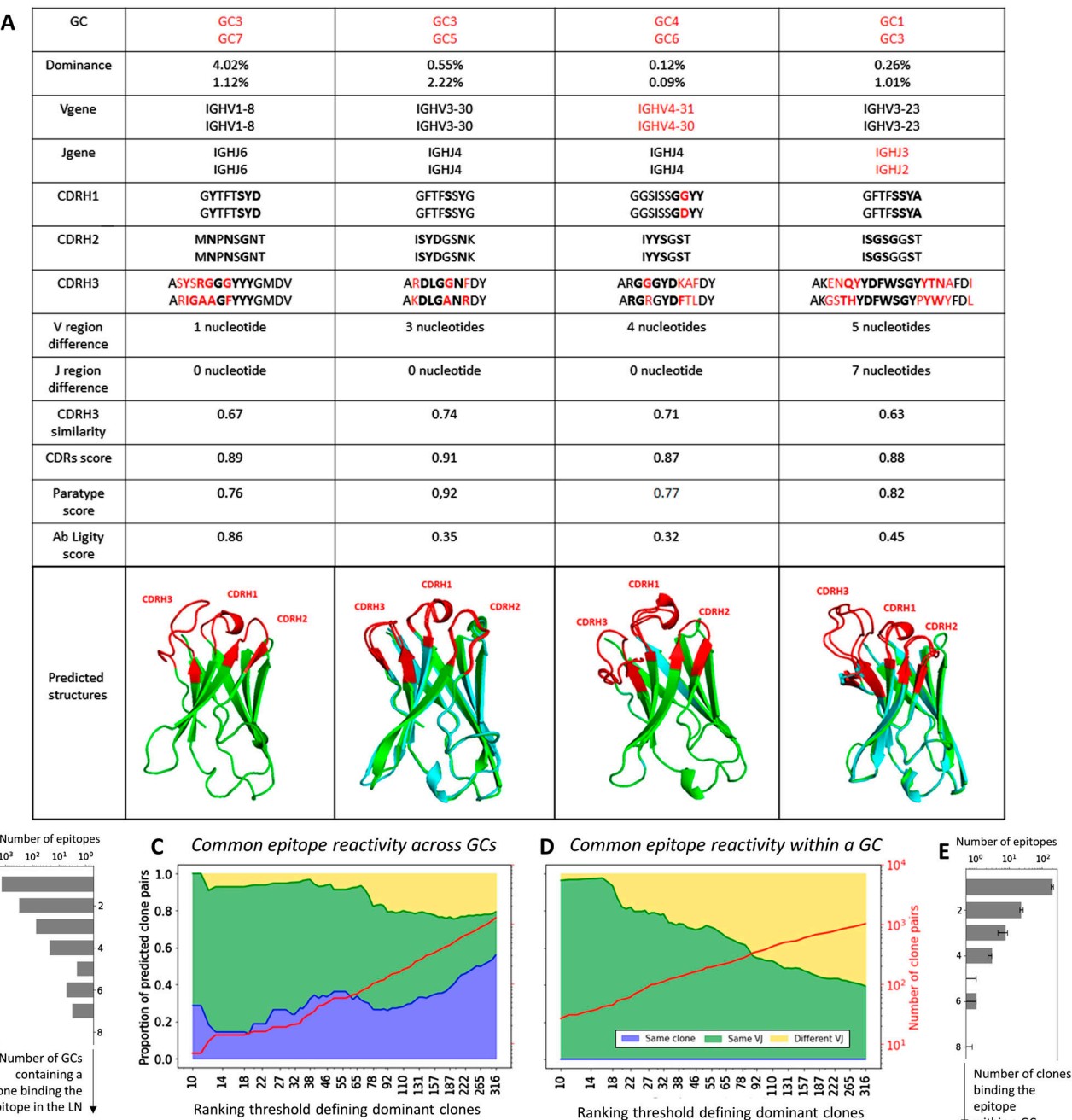

Figure 5. Common epitope reactivity across GCs.
(A) Example of predicted dominant clones pairs that bind to similar epitopes. The paratopes residues are emphasized in bold, and the differences between the two clones are highlighted in red. (B) Number of GCs for which at least one clone is within each structural (epitope binder) group for a dominance threshold of 300. (C, D) Number of predicted clone pairs with common epitope reactivity across different GCs and (D) within the same GC as a function of the dominance ranking threshold. The pairs are separated into three categories: (blue) same clone found in two different GCs, (green) clone with the same V and J gene, but with CDRs similarity <0.84, and (yellow) clone with different V or J genes. (E) Median of the number of clones within each structural (epitope binder) group within each GC for a dominance threshold of 300. The analysis was performed independently for each GC, and the error bar represents the first and third quartile.

sequences in GC8 are generally closer to the root (unmutated V gene, J gene, and consensus CDR3) than GC1 in terms of sequence similarity (Fig 4E), suggesting that GC8 yielded a clone that initiated GC1. This apparent temporal separation (Fig 4D and E) was also observed in the Brainbow mouse model after 20 d of subcutaneous immunization with alum-adsorbed CGG (3, 39), suggesting that mutations may be selected through affinity maturation at different rates and through different evolutionary processes in different GCs. Such a clonal tree pattern was found to different extents in other shared clones and is thus not anecdotal (other shared clones are provided in Fig S14).

### Convergent epitope reactivity across GCs

We studied the structural similarity of dominant clones by considering their paratope structure via paratope modeling. To study the paratope similarity of dominant clones, we assigned to each clone in each GC a representative sequence, defined as the most abundant sequence within that clone. Then, we combined three metrics, the *CDRs similarity*, *Paratype* (40), and *Ab-Ligity* (41), to identify clones that are likely to bind similar epitopes from their heavy chain sequences (Section 6 in Supplemental Data 1). We refer to those predicted clones as clone pairs. If the similarity of the clone pairs is above the optimized threshold for two of the three aforementioned metrics, the clone pair is predicted to have evolved towards binding the same epitope. We show a few illustrative examples of predicted clone pairs in Fig 5A. Although all pairs have a relatively low CDR3 sequence similarity (ranging from 0.63 to 0.74), the pairs exhibit identical (or nearly identical) CDR1 and CDR2, and similar paratope (>0.76) and high structure similarity. The first two clone pairs in the Fig 5A (first and second column starting from the left) could be the same clone misclassified as a different clone because of too many newly accumulated mutations in the CDR3. Alternatively, they could have originated from highly similar naive B cells. The fact that they exhibit differences in the nucleotide sequence of the N region of the V and J genes suggests that they are different clones exhibiting common epitope reactivity. The last two clone pairs (third and fourth column starting from the left) have different V and J genes, respectively, with a 4(7) nucleotides difference between the two V(J) regions and considerable CDR3 dissimilarity, but their paratope and CDR3 loop structure were predicted to be highly similar (Fig 5A).

We classified the predicted clone pairs with common epitope reactivity within one GC and across GCs into the following categories:

- Same clone present in different GCs: this is the most straight forward way to study GC convergent evolution (blue curve in Fig 5C).
- Same V and J gene, but CDR3 similarity <0.84: This category needs to be carefully analyzed because some clones could be misclassified as different because they have accumulated too many mutations in the CDR3 and their CDR3 similarity is thus below 84%, the threshold we used for clone identification in our study (green curve on Fig 5C and D).
- Different V or J gene: This category is based on literature showing that different gene rearrangements can yield convergent epitope

reactivity (40, 42). For this category, however, there is a chance that some pairs are misclassified because of a misalignment in the V or J genes, which is especially likely to happen when their CDR3s are nearly identical (yellow curve on Fig 5C and D).

We also investigated the number of predicted clone pairs within and across GCs based on each of these categories. We considered multiple dominance ranking thresholds, ranging from the top 10 to the top 300 clones in each GC and showed the predicted number of identified clone pairs in each of the three categories with an increased number of dominant clones (Fig 5C and D). We observe an increased number of predicted clone pairs across and within GCs for all three categories, with the number of identified clone pairs approaching 1,000 when the top 300 dominant clones per GC are considered (red line in Fig 5C and D). Most of the predicted clone pairs come from the second category (same V and J genes and CDR3 similarity <0.84, depicted by the green curve in Fig 5C and D).

Regarding the paratope commonalities between dominant clones across and within GCs, we characterized paratope similarity by studying the paratope distance between clone pairs, defined as (paratype distance + CDRs distance)/2. We show that the distribution of paratope distance is very similar across and within GCs, with only a slightly lower average paratope distance within GCs than across GCs (Section 7 in Supplemental Data 1). This finding suggests that different evolutionary forces can lead to convergent immune responses.

## Discussion

In this study, we developed a method to analyze the IGHV repertoire in single GCs by combining LCM of individual GCs from human LNs with NGS Rep-seq from gDNA. Our study revealed not only the heterogeneity of an ongoing immune response in single GCs, within an LN of a 46-yr-old woman, but also the convergent evolution of different evolutionary spaces, the single GCs. The reproducibility of our method was demonstrated by performing replicate IGHV amplification and NGS analyses. We used a multiplex PCR strategy to amplify VDJ junctions, which may be prone to PCR bias, especially when using a low amount of PCR input. However, our replicate analyses showed excellent concordance. Of note, variegated results were obtained when using ~fivefold less PCR input. These results underscore that our method is minimally affected by PCR bias. Moreover, the use of gDNA in our method offers a better alternative to typical mRNA-based studies, which introduce bias due to differential expression of mRNA between different B-cell types, and at the same time, it allows for the study of non-functional clones.

Similarity and diversity analyzes of samples showed that human GCs are distinct evolutionary spaces governed by the evolutionary pressure of antigen capture and selection driving the output of the GC reaction. In line with previous mouse studies (3, 8, 15), we observed a relatively low sequence similarity between the GCs, and heterogeneous clonal diversity in terms of dominance, evenness, richness, and Shannon entropy between individual GCs. The heterogeneous clonal diversity was independent of the clone

identification methods studied and was not a result of sub-sampling because the differences related to sample size were not significant compared with the diversity variation across GCs. The expanded clones showed commonalities in terms of V, D, J gene usage and CDR3 length. This finding suggests convergent evolution of the expanded clones in different individual GCs.

Analysis of the VH gene repertoire revealed that a few VH genes were found more frequently than in public repertoire data that was obtained using a similar multiplex PCR approach as ours (24), suggesting that these were positively selected during the GC re-action in the studied human LN. Accordingly, the VH gene usage was found to be heterogeneous across GCs.

With the help of phylogenetic tree inference, we showed that there is convergent evolution of different GCs of the same LN by identifying the seeding of GC-experienced (potentially memory B cells) B cells in different preexisting GCs, which is in line with previous studies in rodents and humans (3, 19, 20). Although earlier studies identified the frequent presence of clones that were shared between individual GCs in the same LN, both in mice and humans (3, 19), those studies have sequenced limited numbers of B cells with limited statistical analysis/evaluation. Nevertheless, a more recent study in a mouse model argues that this is a restricted phenomenon under typical immunization-boost regimens, with less than 10% clones found in multiple GCs, and secondary GCs consisting predominantly of B cells without prior GC experience (20). Although coming from on ongoing human GC response in an uncontrolled environment, our results are in line with the most recent mouse model studies because we identify ~10% of clones as being shared between GCs.

Using antibody modeling and paratope prediction tools, we quantified the functional convergence of clones in different GCs by predicting candidates of clone pairs that potentially bind the same epitope, suggesting a shared pool of antigens across different GCs. Interestingly, these cases were relatively rare, which suggests that the number of epitopes involved in the LN reaction is high (esti-mated to range from 500 to 5,000 epitopes). In addition, the pre-diction of common epitope reactivity can be used to predict the number of epitopes in the GC and/or the LN. We clustered all clones within structural groups based on the predicted epitope binding, such that each clone in a given group is predicted to bind the same epitope as at least one other clone of the same group. Overall, only a minority of clones were found to bind the same epitope across GCs (Fig 5B and E), thus we expect the number of epitopes present in GCs and LNs to be quite high. With a simple statistical model (Section 8 in Supplemental Data 1), we derived the number of epitopes targeted in the lymph node to be around 5,000, and each GC to be specialized to ~1,000 epitopes on average (Fig S8A and B). These estimations should be taken as a higher bound, as we expect to have missed a large fraction of antibodies that react to the same epitope (recall estimated to be as low as 10%, see Section 6 in Supplemental Data 1). Assuming that 90% of the pairs were missed, our estimate would be significantly reduced to roughly ~500 epi-topes in the lymph node and 100 epitopes in each GC. Given that epitopes are around 15 AAs long (43), and that antigen chains typically consist of hundreds of AAs, many of these epitopes could belong to a single antigen. As an example, insulin was reported to have more than 100 epitopes (44).

Finally, the use of gDNA, as opposed to RNA, which is common practice, allowed us to study and quantify the role of selection in the SHM spectrum by studying the origin and fate of non-functional alleles at the individual GC scale. We showed that the SHM pattern in terms of both position and spectrum was independent of GC B-cell selection. For the first time, we analyzed the non-functional alleles stemming from the SHM or V(D)J recombination process in individual GCs and we observed that the source of the non-functional alleles was approxi-mately equally divided between these two processes. The distribution was roughly consistent across GCs. We identified several mutations that were introduced by SHM and were either selected through affinity maturation or crippled the BCR. Although this type of analysis was previously performed in mouse models in naive and GC B cells from the spleen and Peyer's patches at the B-cell population level, it was not in individual GCs (15). Interestingly, most of the clones with crippling mutations used V genes that are known to be implicated in stereotypic rearrangements in autoimmunity and B-cell malignancies (33, 34), such as IGHV4–34, suggesting counter-selection of those B cells in the GCs. Our results are in line with studies in mouse models, which showed that self-reactive GC B cells are counter-selected or inactivated by SHM (35).

Our study underscores the stochastic nature of SHM, which was found to be selection-independent. It also quantifies its role in producing non-functional clones by analyzing the frequency of non-functional sequences derived from SHM. It shows that the heterogeneity of the individual GCs in terms of diversity and clonal expansions is a conserved phenomenon between mouse and human. In addition, even though the individual GCs are hetero-geneous and separate from each other, at the same time, they exhibit convergent features both in mice (3, 8) and humans.

Our results will help improve existing computational models of the GC reaction, and AID in the development of novel computa-tional models by providing more accurate parameters stemming from a real-life situation in a human LN. As GCs and LNs are sto-chastic systems that display a high level of variability even within the same lymph node of the same individual (45), mathematical models have been widely used to deepen our understanding of the cellular and molecular processes characterizing these complex dynamic systems (9, 46, 47). Nevertheless, these models are limited because of the lack of available data to correctly estimate the nu-merous unknown parameters describing cellular and intracellular interactions. Because our study characterizes clonal abundance, clonal diversity, and clonal functionality data in several GCs of the same lymph node, it represents an important step towards the correct parametrization of GC models. In addition, it could also be used to compare the normal human BCR repertoire with data from disease models, with the final goal of potentially identifying shared BCR structures that could originate from similar antigenic stimuli.

Finally, regarding further research, one question of interest would be to which extent our findings in lymph nodes are appli-cable to different human tissues, such as Peyer's patches and tonsils. If the comparison is relevant, our study conclusions could be extrapolated to investigate the GC reaction mechanisms in other similar environments. In addition, the use of mouse models in combination with gDNA could enable the characterization of the origin and fate of non-functional clones in more controlled envi-ronments. Indeed, controlled mouse immunization experiments

enable the in vitro investigation of the antibodies generated as a response to known antigens, which is not possible in human studies, where GC reactions arise in response to unknown antigens.

# Materials and Methods

### IGH next-generation sequencing/Rep-seq

In brief, LCM of the individual GCs from human LN was combined with the extraction of gDNA and an NGS approach to study both the functional and the non-functional clones.

### Tissue preparation

Patient frozen LN tissue from the pathology department of the Amsterdam UMC hospital was used in this study. (LN studied was a cervical LN resected out of a 46-yr-old woman suffering from chronic sialadenitis. Patient material was obtained according to the ethical standards of our institutional medical ethical committee, as well as in agreement with the 1975 Declaration of Helsinki as revised in 1983). The LN was fresh-frozen in liquid nitrogen shortly after surgical resection. Five serial cryosections (10 $\mu$m per cryosection, CryoStar NX-70; Thermo Fisher Scientific) of the LN tissue on PEN slides (1 mm; Zeiss) were used to obtain higher DNA concentrations. Immunostaining was performed to visualize the GCs. The PEN slides were incubated with drops of hematoxylin (KLINPATH; VWR Life Science) for 1 min at RT, after a quick and gentle wash with demi water and tap water, they dried overnight at RT. In parallel, H&E staining of the frozen tissue was performed on normal (TOMO) slides at Amsterdam UMC diagnostics.

### LCM of individual GCs

After tissue preparation, we used the objective 5x of the Leica PALM-LMD6 to draw and laser capture individual GCs from five serial cryosections of the human LN studied. The laser capture was performed five times for each GC in the same 0.2 ml Eppendorf tube.

### gDNA extraction from individual GCs and DNA concentration

2 $\mu$l 1x proteinase K (recombinant PCR grade, 25 mg; Roche) dissolved in 2.5 ml TE (10 mg/ml) and 20 $\mu$l 1x proteinase K lysis buffer 50 mM Tris–HCl (pH 8, 5-Enzo), 100 mM NaCl (Sigma-Aldrich), 1 mM EDTA (Sigma-Aldrich), 0.5% Tween-20 (Bio-Rad), 0.5% NP-40 (Sigma-Aldrich) were added in the tubes used for the LCM and the samples are incubated at 56°C overnight. The proteinase K was inactivated by incubation for 5 min at 95°C. The Qubit dsDNA HS Assay kit (Invitrogen) was used to quantify the extracted gDNA, according to manufacturer's protocol.

### Multiplex-nested PCR protocol

The gDNA extracted from individual GCs was used for the multiplex-nested PCR, consisting of two rounds. Round 1: amplification of the leader genes region (LD). Round 2: amplification of the FR1 region

with primers containing Illumina adapter overhang nucleotide sequences. The input used for round 2 was the round 1 PCR product. We used technical replicates to check the method reproducibility (duplicates of PCRs for the same DNA/GC).

For round 1: PCR amplification of gDNA from individual GCs (2 ng input) was performed with 1 $\mu$l of JH reverse primer (10 $\mu$M, provided by MERCK) and 1.8 $\mu$l of LD forward primer set pools (10 $\mu$M per primer, provided by MERCK) using 25 $\mu$l of Q5 Hot Start Master Mix (2X) (New England Biolabs) for a total volume reaction of 50 $\mu$l. For round 2: PCR amplification of the round 1 PCR product was performed (1 $\mu$l) with 1 $\mu$l of JH reverse primer (10 $\mu$M, provided by MERCK) and 1 $\mu$l of FR1 forward primer set pools (10 $\mu$M per primer, provided by MERCK) using 25 $\mu$l of Q5 Hot Start Master Mix (2X) (New England Biolabs) for a total volume reaction of 50 $\mu$l.

The following PCR program was used for both round 1 and round 2: 5 min at 95°C; three cycles of 5 s at 98°C and 2 min at 72°C; three cycles of 5 s at 98°C, 10 s at 65°C, and 2 min at 72°C; and 25 cycles of 20 s at 98°C, 30 s at 60°C, and 2 min at 72°C; with a final extension cycle of 7 min at 72°C on a Biometra T-ADVANCED PCR machine. Primers and adaptors used in round 1 and round 2 are provided in Fig S10C.

### Round 2 PCR product purification

Electrophoresis of the round 2 multiplex-nested PCR product was performed on a 1% agarose (Molecular grade; Bioline) gel, followed by gel excision of the 400 bp PCR product with Macherey-Nagel PCR clean up and gel extraction kit, according to manufacturer's protocol.

### NGS library preparation and sequencing

The concentration of the gel excised round 2 multiplex-nested PCR product was quantified by using the Qubit dsDNA HS Assay kit (Invitrogen) according to manufacturer's protocol. 100 ng of this PCR product were used for the preparation of the 16S metagenomic sequencing library for the Illumina MiSeq system, according to manufacturer's protocol. More specifically, for the library construction, the Illumina indexes were attached to the amplicon with 8 PCR cycles using Nextera Index Primer. The final purified (AMPure XP beads) product was then quantified using qRT-PCR according to the qRT-PCR Quantification Protocol.

Guide (KAPA Library Quantification Kits for Illumina Sequencing platforms) and qualified using the TapeStation D1000 ScreenTape (Agilent Technologies). The paired-end (2 × 300 bp) sequencing was then performed by Macrogen using the MiSeq platform (Illumina).

### Sample reads preprocessing

Two samples were acquired for each of the 10 GCs. For each sample, the reads were merged with BBMerge (48) (using the recommended command for optimal accuracy bbmerge-auto.sh). We compared the reads obtained from BBMerge with the ones from PEAR (49) (using a quality index of 30) and no significant differences were found. Then, the primers' and adaptors' sequences were removed by using a customized cutadapt script to remove seven primers from the merged reads (50). The quality of the preprocessed reads

was tested by the FASTQC tool available in the Galaxy platform (https://usegalaxy.org). The sequences were transformed to FASTA format. Finally, the alignments to V and J germline sequences, and the identification of CDRs/FWRs and frameshifts were obtained by submitting the reads to the IMGT-High-V-Quest web portal (28) (http://www.imgt.org/HighV-QUEST/search.action).

## Grouping sequences into clones

Following previous conventions (3, 15, 24, 51, 52, 53), sequences were grouped together into clones if they shared the same V, J gene segments and CDR3 length, and more than 84% junction nucleotide sequence identity. The threshold was optimized using the distance to nearest distribution model (22) and a negation table approach (54), as detailed in the Section 3 in Supplemental Data 1. The junction clustering was performed with a hierarchical agglomerative clustering algorithm (26 *Preprint*), previously proven to be effective at identifying clones (27). As our data are especially susceptible to incorrect V-gene assignment, due to part of FRW1 missing in our sequencing data (the first 15–20 residues), clones with different V genes from the same subgroup were merged together when the CDR3 of their most common sequence were identical.

As a way of checking for the consistency of our analysis, we alternately grouped sequences into CDR3 groups irrespectively of their V and J genes, which can be directly performed with the TRIP tool (25), a software framework implemented in R shiny.

## Similarity between samples

To quantify the similarity between samples $X$ and $Y$, we first normalized the number of sequences belonging to each clone by the total number of sequences in that sample such that $|X| = |Y| = 1$. Then, the similarity was defined as the Sørensen–Dice coefficient, known to be more robust than the Jaccard index when some information is missing from the dataset (55)

$$Similarity = \frac{2|X \cap Y|}{|X| + |Y|} = |X \cap Y| = 1 - \sum_{i \in clones} \frac{|x_i - y_i|}{2}, \quad (2)$$

where $x_i$ and $y_i$ are the abundance of clone $i$ in sample $X$ and $Y$, respectively. A similarity of one indicates perfectly identical samples and one of zero that they do not share any clone.

## Quantifying clonal diversity

To quantify the diversity of GCs, we used different statistical representations of GC clonal composition: richness, dominance, entropy, and evenness (53). The diversity indices are denoted with the unified framework of Hill ($^q D$) (56), where values of $q$ that are smaller than unity disproportionately favor rare species (thus more sensitive to singletons), whereas values greater than unity disproportionately favor the most common species.

· Dominance ($1/^\infty D$): the clonal dominance is defined as the number of cells belonging to the most abundant clone divided by the total number of observed cells in the GC.

· Richness ($^0 D$): the clonal richness is defined as the number of clones in the GC. To provide an estimate of the total number of clones in a GC based on an incomplete sample, we use the bias-corrected Chao1 formula (57, 58):

$$N_{Chao} = N_{obs} + \frac{f_1 \ (f_1 - 1)}{2(f_2 + 1)}, \quad (3)$$

where $N_{obs}$ is the number of observed clones in the GC, $f_1$ is the number of clones detected exactly once, and $f_2$ the number of clones detected exactly twice. In our setup, the second term of Eq. (3) typically equals between 50% and 100% of the number of observed clones.

· Shannon entropy ($\log[^1 D]$): the Shannon entropy has been a popular diversity index in the ecological literature. It is a measure of diversity that accounts *"fairly"* for the abundance of both rare and frequent clones (59) and is thus less affected by the presence of singletons. Defining the sample size $n$, and $p_i = \frac{x_i}{n}$ as the normalized occurrence of clone $i$ in the sample, the Shannon entropy is defined as:

$$H = - \sum_{i \in clones} p_i \ln(p_i). \quad (4)$$

Similar to the Chao richness, we also use an estimator to infer Shannon entropy with incomplete sample information (60). Defining the sample coverage $C = 1 - \frac{f_1}{n}$, we adjust the relative species abundances with $p_j = C$,

$$H_{Chao} = - \sum_{i=1}^{N_{obs}} \frac{p_j \ \log(p_j)}{1 - (1 - p_j)^n}. \quad (5)$$

· Evenness ($^1 D/^0 D$): the evenness quantifies the homogeneity of clonal abundances in a GC. It is defined as the exponential of the Shannon entropy normalized by the richness of the GC (56). The Chao-corrected richness and Shannon entropy were used to compute this indicator:

$$Evenness = \frac{\exp(H_{Chao})}{N_{Chao}}. \quad (6)$$

The evenness is bounded between 0 and 1, where an evenness of one corresponds to a perfectly homogeneous sample, that is, all clones have the same occurrence. For these calculation, we used the Python library diversity (53) (https://github.com/Aurelien-Pelissier/cdiversity).

## Classifying non-functional and functional clones

Each sequence was labeled either as non-functional when having a frameshift or a stop codon or functional otherwise. For each clone, a productivity index, defined as the number of productive sequences

divided by the total number of sequences in that clone (thus between 0 and 1), was computed.

We further classified non-functional sequences into three categories: (i) out of frame, (ii) early stop codon induced by VDJ recombination, and (iii) early stop codon induced by SHM. Out of frame sequences were defined as having an out of frame junction, with a frameshift detected by IMGT. In frame non-functional sequences were assumed to arise either from V(D)J recombination when having at least one stop codon in the IMGT N region or from SHM otherwise.

### Phylogenetic tree representation

We used phylogenetic trees to visualize the evolution of the antibody receptor sequences. In such representation, each founder cell defines the unmutated germline of a new tree, and newly acquired mutations are ideally represented as downstream nodes. We defined the root by taking the unmutated V, J germline and filling the remaining junction region with the consensus sequence of all available sequences within the clone. To compute the trees, the grouped sequences from each clone were first aligned with ClustalW (61). This alignment was necessary to infer a mutation matrix, as tree inference algorithms typically require sequences to be aligned and of the same length, whereas experimentally determined sequences have different lengths and include insertions and deletions. To infer trees from a large amount of sequences, we used a hierarchical clustering approach, where the most similar sequences were grouped together and progressively aggregated with other groups until the root node (maximum distance) was reached (neighbour joining method in ClustalW). Such a method was preferred over typical Bayesian maximum likelihood estimation (39) or Markov chain Monte Carlo sampling (62) because the latter were intractable for the amount of sequences per clone in our data (>5,000 sequences) and would thus require significant sub-sampling leading to an important loss of information in our phylogeny analysis.

### Sequence similarity

The Levenshtein distance (63), defined as the minimum number of edits required to transition from one sequence to the other, is a common metric to quantify sequence similarity. To reduce the bias caused by length differences, we used a normalized Levenshtein distance (64) that incorporates the length of both sequences and satisfies the triangle inequality. Given two strings $s_1$, $s_2$ and $Lev(s_1, s_2)$ the Levenshtein distance between $s_1$ and $s_2$, the normalized Levenshtein distance $Lev_{norm}$ is defined as:

$$Lev_{norm}(s_1, s_2) = \frac{2 \cdot Lev(s_1, s_2)}{|s_1| + |s_2| + Lev(s_1, s_2)}. \tag{7}$$

### Paratope and antibody modeling

The paratope residues were predicted by submitting the antibody sequences to Parapred (65) (https://github.com/eliberis/parapred). To convert the output of Parapred, binding probabilities, into a binary label, we selected a threshold of 0.67, shown to be optimal by the authors of the original article (65). Antibody sequences were submitted to repertoire builder (66) (https://sysimm.org/rep_builder/) to construct the antibody structure from a homology model. As part of the FRW1 region was missing from our sequencing data (the first 15–20 residues), missing nucleotides were filled with the unmutated sequence of their respective IGHV. The obtained structures were visualized with PyMol (67).

### Functional convergence

Although antibodies within the same clones are likely to target a common epitope (40), it has also been observed that clones from different V and J gene backgrounds express common epitope reactivity (68). To identify different clones that are potentially binding to the same target epitope, we combine three approaches.

· CDR similarity: as most of the paratope residues are localized on the CDR loops of the antibody (≈90% according to (69)), antibodies that exhibit high CDR similarity are likely to express common reactivity (although expressing different V or J genes). Given two sequences $s_1$ and $s_2$ with CDRs denoted as H1, H2, and H3, we define the CDR similarity as the average Levenshtein distance between their respective CDRs.

$$CDR\ similarity(s_1, s_2) = \frac{1}{3} \sum_{i=[1,2,3]} Lev_{norm}(Hi_1, Hi_2). \tag{8}$$

The threshold for common reactivity prediction was set to 84% for this metric, as optimized in Section 6 in Supplemental Data 1.

· Paratype (40) simplifies the complex phenomenon of antibody–antigen interaction into sets of shared paratope residues. It was shown to help in identifying antibodies from different clones binding to the same target epitope. Two antibodies are said to be *paratype* if they have a sequence identity across the predicted paratope regions greater than a given threshold. Paratope identity is defined as the number of identical paratope residues (residues that are predicted to be in the paratope in both cases) divided by the smallest number of paratope residues of either sequence being compared. The threshold for common reactivity prediction was set to 76% for this metric, as optimized in Section 6 in Supplemental Data 1.

· Ab-Ligity (41) is a structure-based similarity measure tailored to the antibody–antigen interfaces. Using predicted paratopes on modeled antibody structures, it allows for the identification of sequence-dissimilar antibodies that target highly similar epitopes. In short, the Ab-Ligity framework enumerates all sets of three residues in the paratope structures, and tokenizes them in terms of both the distance between the residues and their intrinsic chemical properties (aliphatic, hydroxyl, sulphur, aromatic, acidic, amine, basic). The obtained set is then compared among other antibodies to obtain a similarity score, and predicted as common epitope if above 0.26, as optimized in Section 6 in Supplemental Data 1.

To find pairs of antibodies that are likely to bind the same epitope, we combine these three metrics. An antibody pair is predicted to bind to a common epitope if it has a similarity above the given threshold for two of the three metrics. In addition,

because a different CDR3 length generally implies a different binding mode, we add the condition that the two antibodies should have the same CDR3 length to be predicted as binding the same epitope. Although some cases of antibodies with different CDR3 length expressing common epitope reactivity were observed (41), such a situation is anecdotal. In fact, removing this constrain considerably reduces the accuracy of the three metrics described above, and existing tools are currently unable to accurately identify such cases, mainly because of the difficulty of modeling the CDR3 loop accurately (70).

### Comparison to public V gene database

We compared the V genes frequencies with a public database of 6 human donors in lymph node (24). LN samples were retrieved form tissues labeled as submandibular lymph nodes (MLN) and inguinal lymph nodes.

### Supplementary materials

The supplementary files attached to this article contains: Section 1 in Supplemental Data 1 and Fig S1: gDNA replicate analysis for different DNA concentration. Section 2 in Supplemental Data 1 and Fig S2: mutations analysis across samples. Section 3 in Supplemental Data 1 and Fig S3: distance to nearest distribution to both negation sequences and sequences within the same sample. Section 4 in Supplemental Data 1 and Fig S4: V, J gene usage combination heatmap. Section 5 in Supplemental Data 1 and Fig S5: common features of the most abundant and shared clones. Section 6 in Supplemental Data 1 and Fig S6: threshold optimization for the Paratype, CDRsim, and Ab-Ligity similarity frameworks. Section 7 in Supplemental Data 1 and Fig S7: distribution of paratope distances for all dominant clones pairs in the same GC and in different GCs. Section 8 in Supplemental Data 1 and Figs S8 and S9: epitope convergence model and epitope diversity in GCs. Fig S10: extensive experimental methods. Fig S11: samples diversity analysis with CDR3. Fig S12: V gene frequency in functional and non-functional clones. Fig S13: Phylogeny reconstruction of some representative F and NF clones. Fig S14: phylogeny reconstruction of some representative shared clones across GCs. Table S1: crippling mutations and stereotyped V genes (excel table).

## Data Availability and Code Availability

Processed sequencing data were deposited on the VDJ server under UUID 8899006209436478995-242ac118-0001-012, publicly accessible at https://vdjserver.org/community/8899006209436478995-242ac118-0001-012. Code availability: the source code to perform convergent antibody binding analysis across repertoires is made available publicly on GitHub at https://github.com/Aurelien-Pelissier/Ab-binding.

## Supplementary Information

## Acknowledgements

The authors thank Kostas Stamatopoulos laboratory for helping with the TRIP tool and for the fruitful discussions. We also thank Thera A M Wormhoudt for assisting in the experimental method development. We thank Nike Claessen for helping with the tissue preparation. This research was supported by the COSMIC European Training Network, funded from the European Union's Horizon 2020 research and innovation program under grant agreement No 765158.

### Author Contributions

A Pelissier: conceptualization, data curation, formal analysis, investigation, methodology, and writing—original draft, review, and editing.
M Stratigopoulou: conceptualization, data curation, investigation, methodology, and writing—original draft, review, and editing.
N Donner: conceptualization and methodology.
E Dimitriadis: data curation and software.
RJ Bende: conceptualization.
JE Guikema: conceptualization, supervision, funding acquisition, methodology, and writing—original draft, review, and editing.
M Rodriguez Martinez: conceptualization, supervision, funding acquisition, and methodology.
CJM van Noesel: conceptualization, supervision, and methodology.

### Conflict of Interest Statement

The authors declare that they have no conflict of interest.

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
