## [Reviewer comments · Life Science Alliance]

Life Science Alliance

Convergent Evolution and B-Cell Recirculation in Germinal Centers in a Human Lymph Node

Aurelien PELISSIER, Maria Stratigopoulou, Naomi Donner, Evangelos Dimitriadis, Richard Bende, Jeroen Guikema, María Rodríguez Martínez, and Carel van Noesel

DOI: <https://doi.org/10.26508/lsa.202301959>

Corresponding author(s): Aurelien PELISSIER, IBM Research - Zurich and Jeroen Guikema, Amsterdam University Medical Centers, Location AMC

Review Timeline:

Submission Date:	2023-01-30
Editorial Decision:	2023-03-27
Revision Received:	2023-06-08
Editorial Decision:	2023-07-25
Revision Received:	2023-07-31
Accepted:	2023-08-02

Scientific Editor: Novella Guidi

Transaction Report:

March 27, 2023

Re: Life Science Alliance manuscript #LSA-2023-01959-T

Carel JM van Noesel
Amsterdam University Medical Centers, Location AMC
Pathology
Meibergdreef 9
Amsterdam 1105 AZ
Netherlands [NL]

Dear Dr. van Noesel,

Thank you for submitting your manuscript entitled "Convergent Evolution and B-Cell Recirculation in Germinal Centers in a Human Lymph Node" to Life Science Alliance. The manuscript was assessed by expert reviewers, whose comments are appended to this letter. We invite you to submit a revised manuscript addressing the Reviewer comments.

Thank you for this interesting contribution to Life Science Alliance. We are looking forward to receiving your revised manuscript.

Sincerely,

B. MANUSCRIPT ORGANIZATION AND FORMATTING:

Reviewer #1 (Comments to the Authors (Required)):

The manuscript by Pelissier et al. entitled "Convergent evolution and B-cell recirculation in Germinal Centers in a human lymph node" describes a very well detailed repertoire DNA sequencing analysis of B-cell homeostasis in a whole lymph node of a healthy individual.

The study is an elegant blend of technical challenges and deeply elaborate bioinformatic analysis. The manuscript details the combined analysis of several topological images of B-cell heavy chain repertoires taken from different locations (germinal centers, GCs) of a single lymph node. This was made possible by combining laser microdissection of 10 different subregions of a lymph node, corresponding to supposedly independent immune responses (GCs), and performing DNA repertoire analysis to obtain 10 large datasets available for further analysis. Beyond the technical challenge, the extensive bioinformatics analysis described in this study allows exploration of a variety of parameters that led to interesting conclusions regarding the diversity of human GC B cells.

The conclusions drawn from this study are that a large "real immune response" observed in a human lymph node is characterized by asynchronous GCs consisting of heterogeneous B cell clones with relatively low sequence similarity. In addition, a small proportion of B cell clones (approximately 10%) is shared between GCs, suggesting that activated B cells can re-engage across GCs. Applying paratope prediction on their repertoire datasets, the authors estimated a functional convergence of clones in different GCs. The authors estimate that the number of targeted epitopes in the lymph node is approximately 5,000, with each GC specialized in approximately 1,000 epitopes; the selection process would divide the number of epitopes by 10, respectively.

The rigorous bioinformatics analysis provided in this study could be used by a wide range of readers for DNA repertoire studies to capitalize on such data sets. By highlighting the conservation between humans and mice of heterogeneity and convergent evolution of individual GSc, the study by Pelissier et al. demonstrates that the development of computational models of the GC content is of major interest for a better understanding of immune responses.

I recommend this manuscript for publication without any change.

Reviewer #2 (Comments to the Authors (Required)):

In this study, Pelissier et al have analyzed in details the IgH repertoire of 10 individual germinal centers from a single lymph node. This cervical lymph node corresponded to chronic inflammation in a patient with sialadenitis. The authors chose to analyze the repertoire based on DNA analysis in order to get access to both functional and non functional Ig sequences with the same efficacy and to avoid biases related to gene expression. They analyzed gDNA from microdissected GCs using multiplex nested PCR followed with MiSeq sequencing.

They now compare their data with those obtained in mice after immunization with well-defined antigens or hapten. As in mice, they show some oligoclonal patterns within GCs, marked with dominant clones, and they identify some shared clones across individual GCs, supporting a phylogenetic tree analysis.

Although the study has merits with the careful description of the repertoire patterns of individuals GCs, it also has a number of limitations which the authors have not tried to fix and which are not sufficiently acknowledged and discussed:

- while the choice of analyzing gDNA is indeed providing a better access to some non-functional sequences and has the advantage of proportionality between the number of B-cells and the number of DNA templates, this approach has a number of weaknesses, which should be dealt with. Major biases in amplification could result from the multiplex PCR and there is no guarantee that the most abundantly amplified V regions correspond to the most abundant templates, since they should simply correspond to a more efficient PCR. It is also quite possible that some V segments are simply lost due to inefficient amplification. The PCR method involves 2x31 cycles followed by 8 cycles for the library construction, which may strongly affect the representation of the initial V region diversity due to biased amplification. Validating this protocol on a series of classical samples (such as PBMCc) and by comparison with a classical RepSeq method based on cDNA and using unique molecular identifiers (UMIs) is necessary before making conclusions about the "most expressed genes".

A second strong limitation is about the claim that convergent clones are identified. CHronic sialadenitis is likely to involve multiple antigens and cannot be expected to be similar to immunization of mice with a single Ag or hapten. Conceptually, it is thus quite uncertain that convergence can be expected. In addition and more importantly, the in silico method used for

evaluating convergence is not validated by any formal proof that "convergent" Abs indeed bind a common Ag, while the prediction omits the light chain structure and is mostly based on CDR3 length and primary structure. To make such predictions more convincing, it would at least be important to show that no convergence is found when the GC RepSeq sequences from the current study are compared with RepSeq data obtained by the same method but with from control tissues or PBMCs from healthy donors, using the same definition of what is supposed to define convergence,

POINT-BY-POINT REPLY TO THE REVIEWERS

Reviewer #1

The manuscript by Pelissier et al. entitled "Convergent evolution and B-cell recirculation in Germinal Centers in a human lymph node" describes a very well detailed repertoire DNA sequencing analysis of B-cell homeostasis in a whole lymph node of a healthy individual. The study is an elegant blend of technical challenges and deeply elaborate bioinformatic analysis. The manuscript details the combined analysis of several topological images of B-cell heavy chain repertoires taken from different locations (germinal centers, GCs) of a single lymph node. This was made possible by combining laser microdissection of 10 different subregions of a lymph node, corresponding to supposedly independent immune responses (GCs), and performing DNA repertoire analysis to obtain 10 large datasets available for further analysis. Beyond the technical challenge, the extensive bioinformatics analysis described in this study allows exploration of a variety of parameters that led to interesting conclusions regarding the diversity of human GC B cells.

The conclusions drawn from this study are that a large "real immune response" observed in a human lymph node is characterized by asynchronous GCs consisting of heterogeneous B cell clones with relatively low sequence similarity. In addition, a small proportion of B cell clones (approximately 10%) is shared between GCs, suggesting that activated B cells can re-engage across GCs. Applying paratope prediction on their repertoire datasets, the authors estimated a functional convergence of clones in different GCs. The authors estimate that the number of targeted epitopes in the lymph node is approximately 5,000, with each GC specialized in approximately 1,000 epitopes; the selection process would divide the number of epitopes by 10, respectively.

The rigorous bioinformatics analysis provided in this study could be used by a wide range of readers for DNA repertoire studies to capitalize on such data sets. By highlighting the conservation between humans and mice of heterogeneity and convergent evolution of individual GSc, the study by Pelissier et al. demonstrates that the development of computational models of the GC content is of major interest for a better understanding of immune responses.

I recommend this manuscript for publication without any change.

Authors' reply: We thank the reviewer for the very positive assessment of our work, and for acknowledging the impact and importance of our study.

Reviewer #2:

In this study, Pelissier et al have analyzed in details the IgH repertoire of 10 individual germinal centers from a single lymph node. This cervical lymph node corresponded to chronic inflammation in a patient with sialadenitis. The authors chose to analyze the repertoire based on DNA analysis in order to get access to both functional and non functional Ig sequences with the same efficacy and to avoid biases related to gene expression. They analyzed gDNA from microdissected GCs using multiplex nested PCR followed with MiSeq sequencing.

They now compare their data with those obtained in mice after immunization with well-defined antigens or haptens. As in mice, they show some oligoclonal patterns within GCs, marked with dominant clones, and they identify some shared clones across individual GCs, supporting a phylogenetic tree analysis.

Although the study has merits with the careful description of the repertoire patterns of individuals GCs, it also has a number of limitations which the authors have not tried to fix and which are not sufficiently acknowledged and discussed:

1) While the choice of analyzing gDNA is indeed providing a better access to some non-functional sequences and has the advantage of proportionality between the number of B-cells and the number of DNA templates, this approach has a number of weaknesses, which should be dealt with. Major biases in amplification could result from the multiplex PCR and there is no guarantee that the most abundantly amplified V regions correspond to the most abundant templates, since they should simply correspond to a more efficient PCR. It is also quite possible that some V segments are simply lost due to inefficient amplification. The PCR method involves 2x31 cycles followed by 8 cycles for the library construction, which may strongly affect the representation of the initial V region diversity due to biased amplification. Validating this protocol on a series of classical samples (such as PBMCs) and by comparison with a classical RepSeq method based on cDNA and using unique molecular identifiers (UMIs) is necessary before making conclusions about the "most expressed genes".

Authors' reply: We thank the reviewer for the thoughtful comments. We agree with the reviewer that any amplification strategy to isolate and characterize VDJ sequences (including ours) may be inherently flawed by amplification biases. We would like to point out that in our strategy we made use of the 'gold-standard' multiplex PCR approach using the established BIOMED-2 primer set [van Dongen *et al.* Leukemia 2003, PMID: 14671650], which was carefully designed to minimize amplification biases, and which is used worldwide to assess clonality and to determine clonal composition of samples. For example, in our study we compare the relative abundance (frequency) of several overrepresented VH genes (IGHV1-2, IGHV2-5 and IGHV1-18) in our dataset to a previously published dataset from lymph nodes and bone marrow obtained from healthy donors [Reference no. 23: Meng *et al.* Nat Biotechnology 2017, PMID: 28829438]. Importantly, these VH genes were not found expanded in that study, whereas a similar PCR and sequencing approach was used, using a similar BIOMED-2-based multiplex PCR as ours. We now specifically mention this in the revised version of the manuscript (line 322, indicated in red textcolor). These results underscore that the analysis of VH gene abundance using this approach yields meaningful results, where potential PCR bias has a minor impact.

Moreover, as of yet there is no primers-based approach to circumvent the potential PCR bias problem. Methods involving the rapid amplification of cDNA ends (RACE) were claimed to not suffer from amplification biases as it does not involve specific primers, but use cDNA as the starting material, which we specifically opted not to use in this study, as our analysis includes the fate of non-functional sequences in a germinal center response. Moreover, we aimed to specifically address the variability in B-cell clones participating in individual germinal centers, and since there is an apparent discrepancy between the number of VDJ RNA molecules/PCR templates and B-cell numbers (which may differ up to a 100-fold), the use of any cDNA-based approach is less suitable to assess the clonal make up of germinal centers in our opinion.

Amplification bias may be particularly problematic when using limited amounts of PCR substrate, yielding variegated results in replicate analyses. However, our approach shows excellent concordance in a replicate setting (**see Figure 1B**). Repeated analysis (amplification and sequencing) on 5 serial tissue sections (which we used in our analyses reported in the manuscript) shows high Dice similarity indices. For comparison, we have now performed an additional replicate analysis using only 1 isolated tissue section per germinal center (so approximately 5-fold less PCR input material), which resulted in much lower Sørensen-Dice similarity indices, indicating variegated bias in each individual amplification/sequence run when PCR substrate is limiting (**See Figure below, right hand panel. Compare to Figure 1B, included here for comparison**). We have included this analysis as **Supplementary Figure 1** in the revised version of the manuscript. We mention this analysis and the implications in the Results section (lines 74-81) and Discussion section (lines 305-308) of the revised version (indicated in red textcolor). Importantly, these data show that although the amplification strategy may suffer from inherent biases, at least these biases appear to be reproducible in our setting. Therefore, there is no ground to assume that the amplification bias differs

significantly between the analyses of the individual germinal centers that we isolated. We do agree, however, that we might have missed certain VH genes in our analyses, but those are mostly likely to be low abundant VH genes. We would like to point out that the primary focus in our analyses is on the *most* abundant VH genes (expanded clones), which are far less likely to be significantly affected by PCR bias. Moreover, in contrast to what was stated by the reviewer, we did not use the term 'most expressed genes' in our manuscript, but rather refer to '*most abundant VH genes encountered*' in our analyses (line 122, line 129-130 of the revised manuscript, indicated in red textcolor).

We appreciate the suggestion to validate our repertoire sequencing of single germinal center using PBMCs, but in our opinion this is not an appropriate approach as the level of clonal expansions in PBMCs is several orders of magnitude lower than in a clonally restricted setting of individual germinal centers. Moreover, it is unclear which criteria should be met from the analysis of PBMCs in order to be satisfactory and applicable to our analyses of single isolated germinal centers. The fact that we find relatively little overlap in the VH genes and clonal representation between individual germinal centers strengthens our confidence in the analyses presented in this manuscript.

The use of an UMI based approach would be helpful if our goal was to most accurately count the actual number of clones and clonal expansions. Rather, our analysis was geared towards comparing *relative abundance* (frequency) of VH genes in germinal centers found in a single human lymph node. Our data clearly show that the clonal representation differs between individual germinal centers, whereas at the same time interesting commonalities can be observed.

Moreover, the limited availability of source material and resources makes the comparison between a cDNA-based UMI and our gDNA-based approach not feasible, as it basically entails that we would have to redo the entire analysis. Although interesting, such technical aspects where different repertoire sequencing approaches are compared in a real-life setting go beyond the scope of the current manuscript.

Figure 1B

Suppl Figure 1

Here we show that the method's performance depends on the DNA concentration. In the samples from the second tissue the Dice similarity is on average lower compared to the ones coming from serial sections (left figure). In fact, we used less tissue in the second example so less DNA isolated, thus making the replicate analysis not accurate. We have included this

analysis as Suppl Figure 1 the revised manuscript, and refer to the results of this analysis in the Results section (lines 74-81).

REVIEWER #2

2) A second strong limitation is about the claim that convergent clones are identified. CHronic sialadenitis is likely to involve multiple antigens and cannot be expected to be similar to immunization of mice with a single Ag or hapten. Conceptually, it is thus quite uncertain that convergence can be expected.

Authors' reply: We agree with the reviewer that ongoing germinal center responses in the lymph node of a chronic sialadenitis likely involves multiple antigens. However, we cordially disagree with the statement that based on this notion convergence can not be expected. In support, we have previously shown that different germinal centers in lymph nodes obtained from patients with similar conditions may harbor cells of similar clonal origin and cells that share stereotypic features [Reference no. 19: Bende *et al.* J. Exp. Med. 2007, PMID: 17938234]. In agreement, in our current analysis we identified clones from different germinal centers that have overlapping CDR3 sequences but exhibit differences in the N-regions, suggesting they originate from different naive B cells. In addition, using paratope modeling we find commonalities in dominant clones across germinal centers.

In our study we estimated the functional convergence of clones in different germinal centers by antibody modeling and paratope prediction tools (where we clearly indicate the upper and lower bounds of our estimations), and show that it is to be expected that only a minority of clones will bind the same epitope across germinal centers, which we clearly indicated in the manuscript. In our honest opinion, to dismiss the whole concept of convergence based on the mere fact that is '*uncertain to be expected*' as stated by the reviewer, does not hold sufficient scientific ground as this is exactly the question that we attempted to answer here.

REVIEWER #2

3) In addition and more importantly, the in Silico method used for evaluating convergence is not validated by any formal proof that "convergent" Abs indeed bind a common Ag, while the prediction omits the light chain structure and is mostly based on CDR3 length and primary structure. To make such predictions more convincing, it would at least be important to show that no convergence is found when the GC RepSeq sequences from the current study are compared with RepSeq data obtained by the same method but with from control tissues or PBMCs from healthy donors, using the same definition of what is supposed to define convergence

Authors' reply: The predictions on structural convergence presented in our study are based on previously published methods that have been rigorously benchmarked on actual antibody binding measurement data [References no. 40 & 41: Richardson *et al.* MAbs 2021, PMID: 33427589 & Wong *et al.* MAbs 2021, PMID: 334482]. In these two studies it was shown that for the accurate identification of common binders the sequences of the IG heavy chains are sufficient. We performed an additional analysis to identify clone pairs based on the Levenshtein distance between CDR3 sequences (which we dubbed '*CDR3sim*'). In the supplementary section 5 we describe the performances of the three different approaches, using a dataset of Pertussis toxin (PTx) binders, which contains heavy chain sequences coding for ~1300 antibodies from mice immunized with PTx. In this dataset, sequences were annotated as PTx-binding or PTx-non-binding based on homogeneous time resolved fluorescence (HTRF) and surface plasmon resonance (SPR) obtained from recombinantly produced antibodies. Using a cross-validation approach we show that using any single method yields a considerable number of false positives, whereas the combination of the three methods dramatically increased the performance.

In our analysis we studied the structural similarities of the most abundant ('dominant') clones in individual germinal centers, which is based on the combination of the three metrics mentioned above (*CDR3sim*, *Paratype* and *Ab-Ligity*) to identify clone pairs. Since the level of clonal expansion is several orders of magnitude lower in PBMNCs (consisting of a large fraction of antigen-naive B cells), this same method will not yield meaningful results in that context. Moreover, it is unclear which benchmark should be met in such an analysis to be applicable or relevant to our analysis of expanded clones in germinal centers. It is not clear to us what the reviewer refers to with 'control tissue', and how this would make our predictions more convincing. As a matter of fact, it was shown that stereotypic B cell receptor rearrangements can be found in the normal repertoire in healthy individuals, but these are typically low abundant and found *across* individuals and not related to expanded clones *within* individuals [Muggen *et al.* Immun. Aging 2019, PMID: 31485252].

July 25, 2023

RE: Life Science Alliance Manuscript #LSA-2023-01959-TR

Carel JM van Noesel
Amsterdam University Medical Centers, Location AMC
Pathology
Meibergdreef 9
Amsterdam 1105 AZ
Netherlands

Dear Dr. van Noesel,

Thank you for submitting your revised manuscript entitled "Convergent Evolution and B-Cell Recirculation in Germinal Centers in a Human Lymph Node". We would be happy to publish your paper in Life Science Alliance pending final revisions necessary to meet our formatting guidelines.

- please upload your main manuscript text as an editable doc file;
- please upload all figure files as individual ones, including the supplementary figure files; all figure legends should only appear in the main manuscript file
- remove figures from the main manuscript text
- please add ORCID ID for the corresponding author--you should have received instructions on how to do so
- please add the Twitter handle of your host institute/organization as well as your own or/and one of the authors in our system
- please add your main, supplementary figure, and table legends to the main manuscript text after the references section;
- please ensure that all authors are added in the Authors Contribution section and that their initials are spelled correctly.
- include callouts for all supplementary figures in your manuscript text.
- please add callouts for Figures 2A and B to your main manuscript text;
- there is no mention that the patient provided consent to use their tissue for research purposes, please provide this when describing the patient tissue in the Materials and Methods section

A. FINAL FILES:

B. MANUSCRIPT ORGANIZATION AND FORMATTING:

Sincerely,

Reviewer #2 (Comments to the Authors (Required)):

I thank the authors for addressing all my comments. I now find the revised version of the manuscript satisfactory for publication.

August 2, 2023

RE: Life Science Alliance Manuscript #LSA-2023-01959-TRR

Mr. Aurelien PELISSIER
IBM Research - Zurich
Bellariastrasse 62
Zurich, Zurich 8038
Switzerland

Dear Dr. Pelissier,

Thank you for submitting your Research Article entitled "Convergent Evolution and B-Cell Recirculation in Germinal Centers in a Human Lymph Node". It is a pleasure to let you know that your manuscript is now accepted for publication in Life Science Alliance. Congratulations on this interesting work.

DISTRIBUTION OF MATERIALS:

Again, congratulations on a very nice paper. I hope you found the review process to be constructive and are pleased with how the manuscript was handled editorially. We look forward to future exciting submissions from your lab.

Sincerely,
